# Lipid ROS- and Iron-Dependent Ferroptotic Cell Death in Unicellular Algae *Chlamydomonas reinhardtii*

**DOI:** 10.3390/cells12040553

**Published:** 2023-02-08

**Authors:** Ramachandran Srinivasan, Hyo-Shim Han, Parthiban Subramanian, Anbazhagan Mageswari, Seong-Hoon Kim, Srikanth Tirumani, Vaibhav Kumar Maurya, Gothandam Kodiveri Muthukaliannan, Mohandass Ramya

**Affiliations:** 1Centre for Ocean Research (DST-FIST Sponsored Centre), MoES-Earth Science and Technology Cell (Marine Biotechnological Studies), Col. Dr. Jeppiaar Research Park, Sathyabama Institute of Science and Technology, Chennai 600 119, Tamil Nadu, India; 2Department of Biotechnology, Sunchon National University, Suncheon 57922, Republic of Korea; 3National Agrobiodiversity Center, National Institute of Agricultural Sciences, Rural Development Administration, Jeonju 54874, Republic of Korea; 4Department of Physiology, Saveetha Dental College & Hospitals, Saveetha Institute of Medical and Technical Sciences (SIMATS), Saveetha University, Chennai 600 077, Tamil Nadu, India; 5PG and Research, Department of Microbiology, Dwaraka Doss Goverdhan Doss Vaishnav College, Arumbakkam, Chennai 600 106, Tamil Nadu, India; 6Agricultural Microbiology Division, National Institute of Agricultural Sciences, Rural Development Administration, Wanju-gun 55365, Republic of Korea; 7Department of Biology, Centre for Structural and Functional Genomics, Concordia University, Montreal, QC H4B 1R6, Canada; 8Division of Food Technology, Cytogene Research & Development, Lucknow 226 021, Uttar Pradesh, India; 9Department of Biotechnology, School of Bio-Sciences and Technology, Vellore Institute of Technology, Vellore 632 014, Tamil Nadu, India; 10Molecular Genetics Laboratory, Department of Genetic Engineering, College of Engineering and Technology, SRM Institute of Science and Technology, SRM Nagar, Kattankulathur 603203, Tamil Nadu, India

**Keywords:** cell death, ferroptosis, glutathione, heme oxygenase, iron, lipid peroxidation

## Abstract

The phenomenon of heat stress leading to ferroptosis-like cell death has recently been observed in bacteria as well as plant cells. Despite recent findings, the evidence of ferroptosis, an iron-dependent cell death remains unknown in microalgae. The present study aimed to investigate if heat shock could induce reactive oxygen species (ROS) and iron-dependent ferroptotic cell death in *Chlamydomonas reinhardtii* in comparison with RSL3-induced ferroptosis. After RSL3 and heat shock (50 °C) treatments with or without inhibitors, *Chlamydomonas* cells were evaluated for cell viability and the induction of ferroptotic biomarkers. Both the heat shock and RSL3 treatment were found to trigger ferroptotic cell death, with hallmarks of glutathione–ascorbic acid depletion, GPX5 downregulation, mitochondrial dysfunction, an increase in cytosolic calcium, ROS production, lipid peroxidation, and intracellular iron accumulation via heme oxygenase-1 activation (HO-1). Interestingly, the cells preincubated with ferroptosis inhibitors (ferrostatin-1 and ciclopirox) significantly reduced RSL3- and heat-induced cell death by preventing the accumulation of Fe^2+^ and lipid ROS. These findings reveal that ferroptotic cell death affects the iron homeostasis and lipid peroxidation metabolism of *Chlamydomonas*, indicating that cell death pathways are evolutionarily conserved among eukaryotes.

## 1. Introduction

Cell death is a fundamental biological process that occurs ubiquitously in all living organisms in response to intrinsic and extrinsic stimuli [1]. In recent decades, several forms of cell death have been established with distinct pathways in varied species, including bacteria, protozoa, and fungi as well as higher plants and humans [2]. Recently, cell death has been broadly classified into accidental cell death and regulated cell death. Compared to accidental cell death, regulated cell death is predominately observed in eukaryotes and involves evolutionarily conserved signaling cascades and distinct pathways [3]. Among these cell deaths, ferroptosis, a new form of cell death, was observed in organisms, which differs physiologically, biochemically, and genetically from apoptosis, necrosis, and autophagic cell death [4]. It is an oxidative, iron-dependent form of cell death associated with cytosolic and lipid ROS [4]. This cell death process can be reversed by iron-chelating agents such as deferoxamine and ciclopirox as well as lipophilic antioxidants such as ferrostatin-1. On the other hand, ferroptosis can be induced by several small molecules such as erastin, sulfasalazine, and sorafenib, which inhibit cysteine/glutamate antiporter (system xc^−^) and reduce the intracellular glutathione content, causing an oxidation–reduction imbalance in cells. RAS-selective lethal 3 (RSL3) directly inactivates GPX4 or depletes glutathione to induce ROS production and lipid peroxidation. The regulatory factors of ferroptosis primarily include the mevalonate pathway, sulfur transfer pathways, and the heat shock factor 1 dependent heat shock protein beta 1 (HSF1-HSPB1) system [5].

In plants, Distefano et al. [6] identified ferroptotic cell death in *Arabidopsis thaliana* during heat stress. Heat stress triggered an iron-dependent cell death that showed hallmarks similar to those observed in mammalian cells such as glutathione and ascorbic acid depletion, lipid peroxidation, and the accumulation of ROS and Fe ions. Moreover, an exogenous addition of iron triggered ferroptosis-mediated cell death in plants infected with viruses and fungi [7]. Recently, Aguilera et al. also observed oxidative, highly regulated, iron-dependent cell death in the response of *Synechocystis* sp. PCC6803 to heat shock [8]. Iron is an important mediating factor for the ROS production by Fenton’s reaction that stimulates lipid peroxidation during ferroptosis [7]. Like plants, microalgae, including *Chlamydomonas*, firmly regulate iron assimilation through the differential expression of various iron assimilatory genes [9]. Though the process of iron acquisition and homeostasis in *Chlamydomonas* is well established, its possible role during abiotic stress is unknown. In microalgal species, apoptosis-like cell death in response to stress is either caspase-dependent or -independent, with established cellular processes such as cell shrinkage, DNA fragmentation, chromatin condensation, the loss of membrane potential, the externalization of phosphatidylserine, and ROS accumulation in cell organelles [10]. These studies indicate that the mode of cell death in microalgae is not precisely defined and appears to be diversified compared to the mode in higher organisms. However, similar to plants and mammalian cells, *Chlamydomonas* have also been found to exhibit defined characteristics of ferroptotic cell death during heat stress, which includes the depletion of the glutathione–ascorbate redox system [11], ROS accumulation, and lipid peroxidation [12]. However, the roles of these key players in iron-dependent cell death in microalgae remain to be examined. Understanding cellular processes such as ferroptosis can help us to understand any unsolved phenomenon observed in algae, provide insight into its biotechnological implications, and even clarify the adaptation mechanisms used by algae in specific ecological niches. The present study was aimed at investigating the underlying mechanisms of the ROS- and iron-mediated signaling cascades of the ferroptotic pathway using *Chlamydomonas reinhardtii* as a model system during heat shock (at 50 °C).

## 2. Materials and Methods

### 2.1. Chalmydomonas and Culture Treatment Conditions

The wild-type strain of *C. reinhardtii* CC-124 was obtained from the *Chlamydomonas* Research Center, University of Minnesota, Minneapolis, MN, USA. The culture was maintained in a tris-acetate phosphate (TAP) medium at 25 °C with continuous illumination of 50 µE m^−2^ s^−1^. For the heat treatment, the exponential-phase cells (2 × 10^6^ cells/mL) were incubated at 50 °C for 10 min in a shaking water bath in the dark. Then, the cells were recovered under continuous light at 25 °C for programmed cell death to occur [13]. The cells were preincubated with the ferroptosis inhibitors (Cayman Chemical, MI, USA) 1 µM ferrostatin-1 (Fer-1), 10 µM ciclopirox (CPX), and 10 µM Ac-DEVD-CHO (CHO) for 24 h before heat shock at 50 °C for 10 min. An RSL3 (10 µM) treatment for 24 h was used as a positive control for ferroptosis induction. Cultures without pretreatment were used as a control. The cell viability was measured by costaining the cells with a 1:1 ratio of 5 µM Syto 9 green and 20 µM propidium iodide (Invitrogen, Thermo Scientific, MA, USA) at 25 °C for 15 min in the dark and washing twice with 1 × phosphate-buffered saline (PBS). The cells were examined and counted under fluorescence microscopy (Leica Microsystems, Wetzlar, Germany) with Ex/Em of 420–470/485–525 nm for Syto 9 green and Ex/Em of 535/617 nm for propidium iodide (PI) [14]. Cell death was calculated as the percentage of dead cells (PI-positive) compared to the total number of cells. The cell counting of each sample was performed from three different randomly chosen microscopic fields containing at least 85–100 cells.

### 2.2. Fluorescence Microscopy and Measurements

After the heat shock treatment (50 °C), cells were stained with 5 µM H_2_DCFDA (Sigma, D6883) for the detection of ROS accumulation, 5 µM C11-BODIPY™ 581/591 (Cayman, 27086) for the detection of lipid peroxidation, 3 µM JC-1 (Cayman, 10009172) for staining the mitochondrial membrane potential, 15 µm Calcium Green-1, AM (Cayman, 20400) for staining cytosolic calcium, and 1 µM FerroOrange (Merck, SCT210) for iron staining at room temperature for 20–30 min, followed by two washes with 1X PBS. Fluorescence images were captured using a Leica fluorescence microscope (Leica Microsystems, Germany) equipped with a Leica DFC9000 sCMOS camera. The excitation/emission of fluorescent dyes used the following filters: H_2_DCFDA (Ex/Em: 492/515 nm), C11-BODIPY™ 581/591 (Ex/Em: 460–495/510–550 nm), JC-1 (Ex/Em: 488/538 nm for monomers and 596 nm for oligomers), Calcium Green-1 (Ex/Em: 506/531 nm), and FerroOrange (Ex/Em: 543/580 nm). For fluorescence intensity, 100 µL of stained cells were transferred into a black 96-well plate and fluorescence was measured using a Synergy HTX Multi-mode Plate Reader (BioTek, VT, USA) with the indicated excitation and emission wavelengths.

### 2.3. RNA Isolation and Real-Time Quantitative PCR (RT-qPCR) Analysis

Total RNA was isolated from control and experimental cultures using Trizol (Invitrogen, Thermo Scientific, MA, USA) and reverse-transcribed using a RevertAid First strand cDNA Synthesis Kit (Thermo Scientific, MA, USA) as per the manufacture’s protocol. RT-qPCR was performed with a CFX96 Touch Real-Time PCR Detection System (Bio-rad, CA, USA) using iTaq universal SYBR Green Supermix. The list of RT-qPCR primers is given in Appendix A. The relative gene expression of the target genes was normalized against CBLP (*Chlamydomonas* G protein ß-subunit-like polypeptide (endogenous control)) using the 2^−ΔΔCT^ method [15]. All the reactions were performed in triplicate with three biological replicates.

### 2.4. Lipid Peroxidation Assay and Lipoxygenase (LOX) Activity

The ferrous oxidation xylenol orange (FOX) assay was performed to determine the levels of lipid hydroperoxide (LOOH). Treated *Chlamydomonas* cells were harvested, weighed, and used for total lipid extraction. The extracted lipids were treated with or without 10 mM triphenyl phosphine (TPP) for 30 min to reduce lipid peroxides. The TPP-treated and untreated lipids were mixed with 0.5 mL of FOX reagent (1:1 ratio of 50 mM xylenol orange and 5 mM iron(II) sulfate heptahydrate) for 30 min at room temperature, and the absorbance was measured at 560 nm [12]. Standards of H_2_O_2_ (0 to 20 µM) were used to create a calibration curve. For LOX activity, *Chlamydomonas* cells were harvested and homogenized with 5 mL of a 50 mM ice-cold potassium phosphate buffer (1% polyvinylpyrolidone and 0.1% triton x-100). The samples were centrifuged at 10,000× *g*, and the supernatant was used for LOX activity, which was carried out according to the protocol of Awad et al. [16]. The LOX activity was detected by monitoring the absorbance of conjugated dienes at 234 nm using a spectrophotometer (UV-1280, Shimadzu, Tokyo, Japan).

### 2.5. Iron Content Measurement and HO-1 Activity

The measurement of the iron content was performed as described in a previous study [17]. *Chlamydomonas* cultures were harvested and homogenized with a 0.1 M phosphate buffer (pH 7.2). From the samples, 100 µL of homogenates were mixed with 0.1 mL of 98% sulfuric acid and 70% nitric acid. Samples were completely digested at 100 °C for 3 h, following which 50 µL of 60% perchloric acid was added, and the mixture was dried completely at 100 °C. Finally, 0.5 mL of sterile distilled water, 0.25 mL of 1% thioglycolic acid, 1.5 mL of saturated sodium acetate, and 1.0 mL of 0.08% bathophenanthroline in ethanol were added to the ash samples and vortexed well. The samples were centrifuged at 12,000× *g*, and the supernatant was used to measure the total iron content colorimetrically at 535 nm. The total iron content (ng mg^−1^ dry weight) was calculated using a standard iron solution (Sigma Aldrich, MO, USA).

For the HO-1 activity, *Chlamydomonas* cells were harvested and homogenized with an ice-cold 50 mM phosphate buffer (pH 7.4) containing 1 mM phenylmethylsulfonyl fluoride, 0.2 mM ethylenediaminetetraacetic acid, and 0.25 mM sucrose. The cell homogenate was centrifuged at 10,000× *g* for 20 min, and the resulting supernatant was assessed for HO-1 activity [18]. The activity of HO-1 was assessed in a 200 µL reaction mixture containing 40 µL of enzyme extract, 0.15 mg mL^−1^ bovine serum albumin, 10 µM hemin, 0.025 units mL^−1^ spinach ferredoxin-NADP^+^ reductase, and 4.2 µM ferredoxin. The reaction mixture was incubated at 37 °C for 1 h, and the enzyme activity was measured at 650 nm using a UV spectrophotometer (UV-1280, Shimadzu, Tokyo, Japan).

### 2.6. Determination of Ascorbate (AsA) and Glutathione (GSH) Contents

The estimation of the total ascorbate (AsA and DHA), reduced ascorbate (AsA), and dehydroascorbate (DHA) was carried out using the method of Gossett et al. [19]. In this method, the reduction of Fe^3+^ to Fe^2+^ with ascorbic acid in an acidic medium resulted in the formation of a red chelate between Fe^2+^, and 2,2′-bipyridyl was read at 525 nm. Dehydroascorbate was determined by subtracting AsA from AsA and DHA. The ascorbate content was calculated using a series of standards of L-ascorbate (Sigma Alrdich, MO, USA). GSH and glutathione disulfide (GSSG) were quantified by following the 5,5′-dithiobis (2-nitrobenzoic acid) method using glutathione reductase and 2-vinylpyridine and were measured at 415 nm according to the protocol of Griffith [20]. The GSH content was measured by subtracting the GSSG content from total glutathione content using the standard curve that was prepared with different concentrations of GSH.

### 2.7. DNA Fragmentation Assay

After the treatment, the algal cells were collected via centrifugation at 5000× *g* for 5 min and washed with 1X PBS. The DNA extraction was carried out using a DNeasy Plant Mini Kit (Qiagen, Hilden, Germany) according to manufacturer’s instructions. The purified DNA (5 µg) was subjected to electrophoresis in a 1.5% agarose gel and stained with ethidium bromide. The bands were visualized and documented with a Gel Documentation System (Bio-Rad, CA, USA).

### 2.8. Statistical Analysis

All experimental procedures were performed using three biological samples (*n* = 3). GraphPad Prism statistical software v5.0 was used to analyze the experimental data using a one-way ANOVA with Bonferroni’s post hoc test.

## 3. Results and Discussion

### 3.1. Ferroptosis Inhibitors Prevent Cell Death Induced by Heat Shock

*Chlamydomonas* is an excellent model system to study the underlying mechanism of the heat stress response [21]. In earlier studies, the heat stress response in *Chlamydomonas* demonstrated the common hallmark reactions of ferroptosis, similar to other eukaryotes, which included the production of reactive oxygen species (ROS), lipid peroxidation, GPX4 downregulation, and GSH and ascorbate depletion [11,12,22]. To investigate ferroptosis-like cell death in *Chlamydomonas*, cells were subjected to RSL3 and a heat shock (50 °C) treatment in the presence or absence of ferroptosis inhibitors, and cell death was observed at regular intervals (Appendix A). The results indicate that the addition of ferroptosis inhibitors (Fer-1 and CPX) significantly reduced the cell death induced by heat shock at 50 °C (Figure 1A). The cell viability was also assessed by staining the cells with Syto9 green and PI (Figure 1B). These findings suggest that Fer-1 and CPX significantly prevented ferroptosis-like cell death in *Chlamydomonas* subjected to heat shock.

### 3.2. Hallmarks of Ferroptosis-like Cell Death in Chlamydomonas during Heat Shock

#### 3.2.1. Heat Shock Induced Cytosolic ROS and Lipid Peroxides

Ferroptosis is an iron-dependent cell death caused by oxidative burst and results in an accumulation of cytosolic and lipid ROS [4,7]. To examine the accumulation of cytosolic and lipid ROS in *Chlamydomonas*, treated cells were stained with the fluorescence probes H2DCFDA and C11-BODIPY 581/591, respectively. The maximum fluorescence intensity was observed in cells treated with RSL3 and heat shock (50 °C). However, cells preincubated with inhibitors before the heat shock (50 °C) and RSL3 treatment showed a significant decrease in the accumulation of cytosolic and lipid ROS (Figure 2A–C). Lipid peroxidation is a chain reaction initiated by hydrogen abstraction or oxygen addition to the acyl groups of polyunsaturated fatty acids (PUFA). During oxidative stress, the biosynthesis of lipid peroxides can be carried out by Fenton-type chemistry or via an enzymatic process, most notably LOX [23]. In order to further investigate lipid peroxidation, the LOOH levels were monitored using a FOX assay, and LOX activity was studied in cells preincubated with inhibitors as well as under heat shock at 50 °C (Figure 3). As shown in Figure 3A, the LOOH content was notably increased in RSL3-treated as well as heat-stressed cells, whereas it was significantly mitigated upon pretreatment with Fer-1 and CPX. Similarly, the LOX activity was found to be significantly higher in the cells treated with RSL3 and the cells subjected to heat shock at 50 °C (Figure 3B). These results are consistent with the previous findings of Legeret et al. [22], where lipidomic analyses revealed that after 60 min at 42 °C, a strong decrease in the polyunsaturated membrane lipids of *Chlamydomonas* was observed. In addition, a transcriptome analysis also showed that there was an upregulation of genes associated with the lipid-degrading signaling pathway, including LOX.

#### 3.2.2. Heat Shock Induced Mitochondrial Dysfunction and Cytosolic Calcium Levels

In the programmed cell death mechanism, mitochondrial dysfunction is associated with the membrane potential and the release of pro-apoptotic protein into the cytosol [24]. Recently, researchers have found the important role of mitochondria in ferroptosis-mediated cell death [25]. In this study, fluorochrome JC-1 was used to assess the mitochondrial membrane potential (MMP) of *Chlamydomonas* using fluorescence microscopy. It forms J-aggregates, which exhibit a red fluorescent signal, when the membrane potential is high, whereas when it remains as a monomer, a green fluorescent signal is observed, denoting a low membrane potential. In cells treated with RSL3 and heat shock (50 °C), the JC-1 fluorescence signal shifted from red to green, suggesting a decrease in the MMP. However, cells preincubated with inhibitors significantly restored the MMP compared to the RSL3 and heat shock treatment (Figure 2A,D). Cytosolic calcium is a key secondary messenger in the stress response and regulated cell death processes [26]. Previous studies reported that ferroptotic cell death was associated with lipid peroxidation and increased calcium fluxes through specific channels [27,28]. In this study, Calcium Green-1, a fluorescent dye, was used to measure the cytosolic calcium levels in *Chlamydomonas* cells subjected to heat shock with or without inhibitors (Figure 2A,E). Upon the heat and RSL3 treatment, cells showed a significant increase in cytosolic calcium, whereas preincubation with inhibitors resulted in a significant decrease in cytosolic calcium. This highlights the involvement of cytosolic calcium during heat-shock-induced cell death in microalgae.

#### 3.2.3. Heat Shock Triggered Fe^2+^ Labile Ferrous Iron Accumulation via Modulating HO-1 Activity

During ferroptosis, the accumulation of intracellular labile iron directly catalyzes the formation of ROS and causes oxidative stress via Fenton reactions, thereby promoting the peroxidation of lipids, proteins, and nucleic acids [29]. To detect labile ferrous iron, cells were stained with the orange fluorescent probe FerroOrange, and the labile ferrous iron content was observed. It was found that the heat- and RSL3-treated cells accumulated high levels of ferrous iron, which were significantly reduced by Fer-1 and CPX, as revealed by decreased orange fluorescence (Figure 2A,F). Similarly, treatment with inhibitors significantly decreased the level of labile ferrous iron upon heat shock in *Chlamydomonas* (Figure 2G). Heme oxygenase-1 (HO-1) is an important rate-limiting enzyme involved in the regulation of intracellular iron homeostasis by the detoxification of free heme. During ferroptotic cell death, the upregulation of HO-1 may become detrimental and cytotoxic due to the accumulation of pro-oxidant labile iron, resulting in ROS production [30]. In plants, HO-1 is induced by oxidative stress stimuli, and the activation of HO-1 gene expression is considered to be an adaptive cellular response to survive exposure to environmental stresses [31]. However, only a few reports have observed that HO-1 is directly involved in regulating heavy-metal-induced ROS stress in microalgae [32,33]. Recent studies have demonstrated that the activation of HO-1 may become detrimental and cytotoxic due to increased intracellular iron arising from iron stores, which induces ferroptosis [34]. The precise mechanism of HO-1 and its iron homeostasis in microalgae in response to environmental cues remains unknown. Therefore, we investigated the transcript level and enzyme activity of HO-1 in response to RSL3 and heat shock (50 °C). The transcript level and enzymatic activity of HO-1 was found to be higher in cells treated with RSL3 and heat shock at 50 °C (Figure 4). This result was consistent with previous findings where the upregulation of HO-1 was significantly increased in *C. reinhardtii* during heavy-metal-induced oxidative stress [32,33]. Similarly, several studies have reported that the expression of HO-1 was significantly increased under oxidative stress in plant and mammalian cells [35,36,37]. However, cells preincubated with inhibitors showed a significant decrease in the transcript level and activity of HO-1 (Figure 4A,B). These results suggest that heat shock induces iron-dependent cell death with an increase in the accumulation of labile ferrous iron by modulating the HO-1 activity of *Chlamydomonas*.

#### 3.2.4. Heat Shock Inactivates Glutathione Peroxidase (GPX) via Glutathione and Ascorbate Depletion

Lipid hydroperoxide glutathione peroxidase (PHGPX) plays a vital role in ferroptosis and is the key regulator of the inhibition of lipid hydroperoxides. PHGPX converts GSH into GSSG and reduces cytotoxic LOOH [38]. The inhibition of PHGPX activity can lead to the accumulation of lipid peroxides, which is a biomarker of ferroptosis. Miao et al. [39] found that the *Chlamydomonas* GPX5 protein showed a conserved active site domain belonging to PHGPX, with similarity to human GPX4. Previous studies reported that GPX5 expression was significantly higher in response to a single oxygen stress induced by photosensitizers and prevented the accumulation of lipid hydroperoxides in *C. reinhardtii* [40,41]. Similarly, a gpx5 mutant strain showed depressed synthesis of lipid droplets compared to parental strain CC4348 [39,42]. In this study, the expression of GPX5 was significantly decreased in RSL3- and heat-treated cells (Figure 5A), which was consistent with a previous report by Muhlhaus et al. [11]. However, there was no significant change in the cells treated with the inhibitors. Likewise, GSH and AsA are also important constituents of the redox regulation in eukaryotic cells, including microalgae. The depletion of GSH and AsA is clearly associated with ROS and the cell death pathway in both plant cells and microalgae. A study of the heat stress response in *Chlamydomonas* using quantitative shotgun proteomics revealed that significant decreases occurred in a few proteins that are related to redox homeostasis, including the thioredoxin system, glutathione metabolism (GPX5), and ascorbate oxidases/reductases [11]. As a result, the levels of GSH and AsA in *Chlamydomonas* cells treated with heat shock at 50 °C in the presence or absence of ferroptosis inhibitors were examined, and the results revealed that the levels of GSH and AsA were significantly lowered in cells under heat shock, whereas cells treated with RSL3 showed no significant change when compared to control cells (Figure 5B–F). RSL3 is a selective inhibitor of GXP4 without GSH depletion. It triggers ROS and lipid peroxidation in plant and mammalian cells [43,44]. Similar to GPX5 expression, none of these inhibitors could prevent the depletion of the GSH and AsA pool, suggesting that the GSH-AsA redox regulatory system is an early event of the ferroptotic pathway but not a consequence of lipid peroxidation. The results obtained in the current study corroborate the findings of earlier reports in plant cells [6]. Overall, these results indicating that heat shock (50 °C) induces oxidative and iron-dependent programmed cell death in *Chlamydomonas* cells, similar to plant and mammalian cells.

#### 3.2.5. Heat-Shock-Induced Cell Death Involves Caspase-like Activity but Does Not Cause DNA Fragmentation

Similar to land plants, *Chlamydomonas* lacks canonical caspases; however, proteases have been shown to be responsible for caspase-like activities in algal cells during cell death [45,46]. To determine the caspase-like activity in *Chlamydomonas* during RSL3 and heat shock treatment, we investigated the effect of preincubation with the reversible caspase-3 inhibitor CHO. We observed that preincubation with CHO significantly reduced the percentage of cell death induced by RSL3 and heat shock (Appendix A). This study suggested that a caspase-like activity was involved in the pathway triggered by RSL3 and heat shock. Programmed cell death, such as apoptosis, has been widely reported in *Chlamydomonas* and appears to share certain morphological characteristics with apoptosis-like cell death in multicellular organisms, including DNA fragmentation [10]. In previous studies, DNA fragmentation in *Chlamydomonas* under heat stress (50 °C for 10 min) was observed only after 14–16 h of treatment but not at different time points after heat treatment [13,47]. To assess apoptosis-like cell death, we performed a DNA fragmentation assay using agarose gel electrophoresis. No DNA fragmentation was observed in *Chlamydomonas* cells for 12 h after the heat shock and RSL3 treatment (Appendix A). However, DNA fragmentation cannot be completely ruled out at the late stage of the cell death pathway, which might be because the majority of cells had lost their viability. Our findings suggest that it might not occur until at least 12 h after heat shock, when cell death has already been detected. These results are consistent with previous findings [6,43] in *Arabidopsis thaliana* cells.

## 4. Conclusions

The present investigation indicates that heat shock (50 °C) induces iron-dependent cell death with hallmarks of GPX5 inactivation through GSH and AsA depletion, which leads to an accumulation of intracellular free iron that generates ROS and lipid peroxidation (Figure 6). These biochemical and physiological changes in *Chlamydomonas* provide evidence of iron-dependent cell death during heat shock in the microalgae, which is well defined in plant as well as mammalian cells. Ferroptosis inhibitors such as Fer-1 and CPX rescued *Chlamydomonas* cells from ferroptotic cell death during heat shock (50 °C). The occurrence of ferroptosis-mediated cell death in *Chlamydomonas* provides new insights into the evolutionarily conserved mechanisms of cell survival and programmed cell death in eukaryotic cells under stress conditions. Moreover, understanding the finer aspects of ferroptotic cell death regulation in microalgae could be useful for the effective control and management of several natural and anthropogenically induced phenomena. To give an example, such knowledge could help us effectively manage harmful algal blooms caused by climate change and their effects on ecosystems.

## Figures and Tables

**Figure 1 cells-12-00553-f001:**
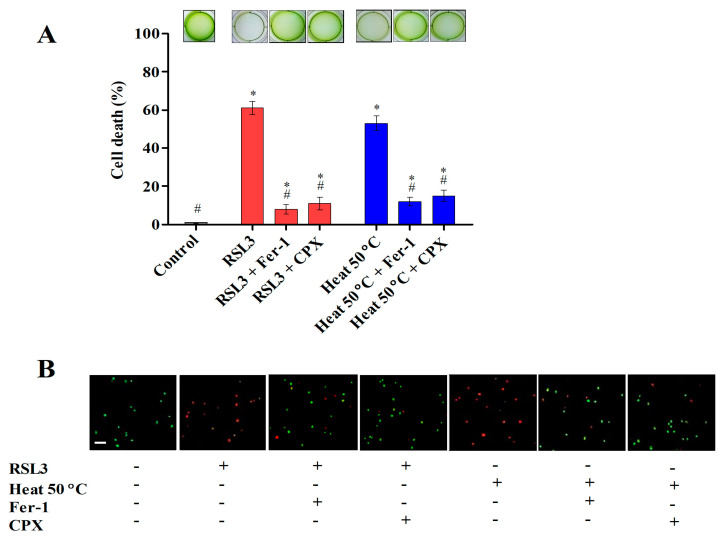
Ferroptosis inhibitors prevent RSL3- and heat-shock-induced cell death in *Chlamydomonas*. (**A**) The viability of *Chlamydomonas* cells exposed to RSL3 and heat stress at 50 °C for 10 min after preincubation with ferroptosis inhibitors (1 µM Fer-1 and 10 µM CPX) for 24 h in six-well plates. (**B**) Live (green) and dead (red) cells were detected using Syto9 green and PI under fluorescence microscopy. The data represent means ± SDs (*n* = 3). * indicates significant differences with respect to the untreated control. # indicates significant differences with respect to the RSL3 or 50 °C heat treatments (*p* < 0.05). Scale bar = 50 µm.

**Figure 2 cells-12-00553-f002:**
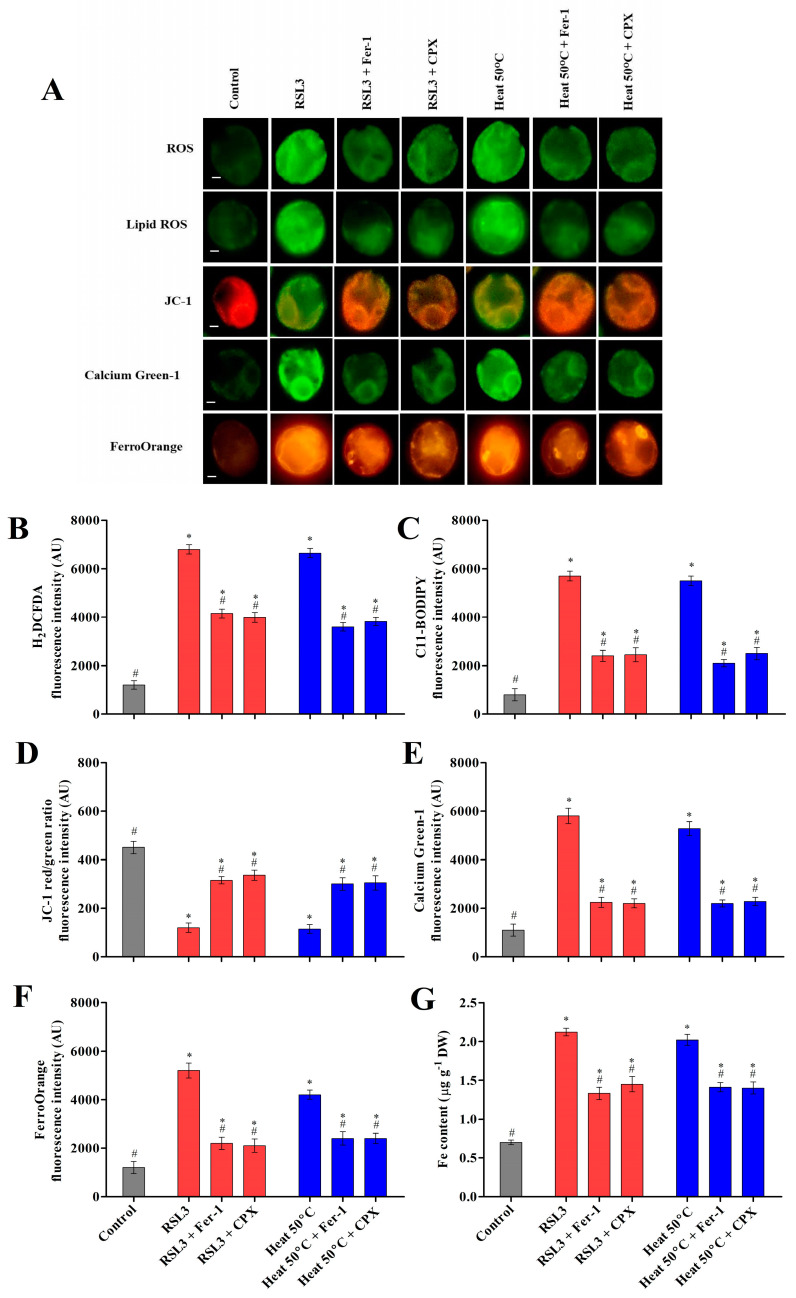
Characteristics and biomarkers of ferroptotic cell death in *Chlamydomonas* after 1 h and 3 h of RSL3 and heat shock (50 °C) treatment. (**A**) Representative images of *Chlamydomonas* cells stained by ferroptotic biomarkers (H_2_DCFDA, C11-BODIPY, JC-1, Calcium Green-1, and FerroOrange) and visualized using fluorescence microscopy. The fluorescence intensity of ROS accumulation was determined using H_2_DCFDA (**B**). The production of lipid peroxidation was determined using C11-BODIPY (**C**). Mitochondrial dysfunction was determined using the red/green fluorescence ratio of JC-1 (596/538 nm) (**D**). The intracellular calcium level was quantified using Calcium Green-1 (**E**). The accumulation of Fe^2+^ ions was measured using FerroOrange (**F**) and the quantification of the iron content (**G**). The data represent means ± SDs (*n* = 3). * indicates significant differences with respect to the untreated control. # indicates significant differences with respect to the RSL3 or 50 °C heat treatments (*p* < 0.05). Scale bar = 10 µm.

**Figure 3 cells-12-00553-f003:**
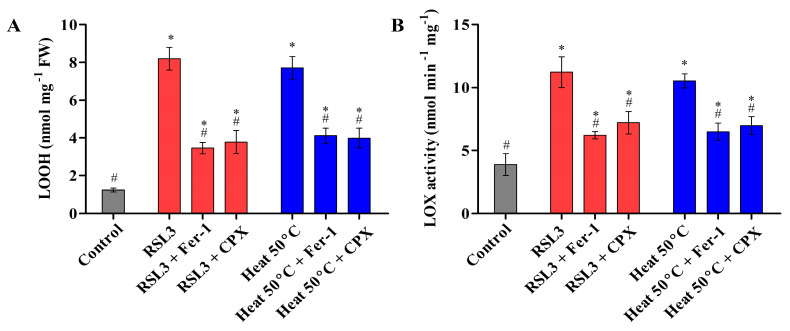
LOOH level quantification using the FOX assay (**A**) and LOX enzymatic activity (**B**) of *Chlamydomonas* cells treated with RSL3 and heat shock (50 °C). The data represent means ± SDs (*n* = 3). * indicates significant differences with respect to the untreated control. # indicates significant differences with respect to the RSL3 or 50 °C heat treatments (*p* < 0.05).

**Figure 4 cells-12-00553-f004:**
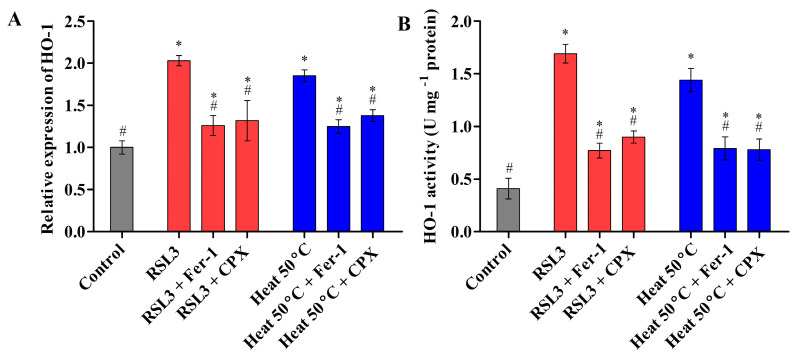
Transcript level (**A**) and enzyme activity (**B**) of HO-1 in *Chlamydomonas* cells after RSL3 and heat shock (50 °C) treatments. The data represent means ± SDs (*n* = 3). * indicates significant differences with respect to the untreated control. # indicates significant differences with respect to the RSL3 or 50 °C heat treatments (*p* < 0.05).

**Figure 5 cells-12-00553-f005:**
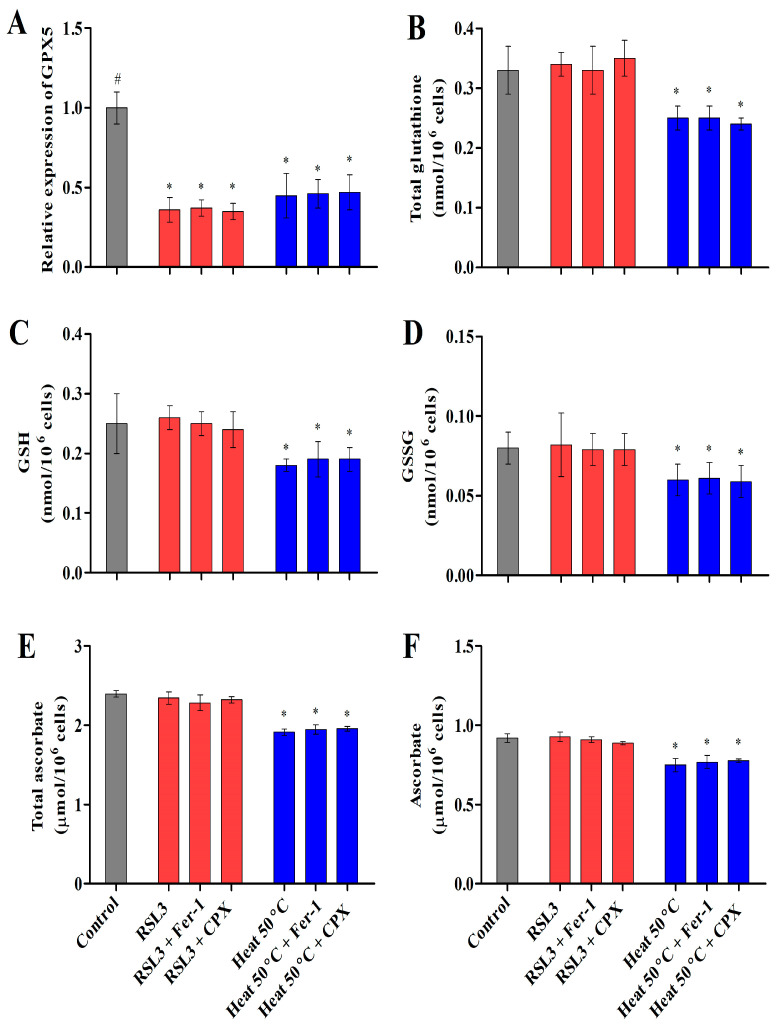
Evidence of ferroptosis involves the inactivation of the GXP5 transcript level (**A**) and the depletion of total glutathione (**B**), GSH (**C**), GSSG (**D**), total ascorbate (**E**), and ascorbate (**F**) in *Chlamydomonas* under RSL3 treatment and heat shock (50 °C). The data represent means ± SDs (*n* = 3). * indicates significant differences with respect to the untreated control. # indicates significant differences with respect to the RSL3 or 50 °C heat treatments (*p* < 0.05).

**Figure 6 cells-12-00553-f006:**
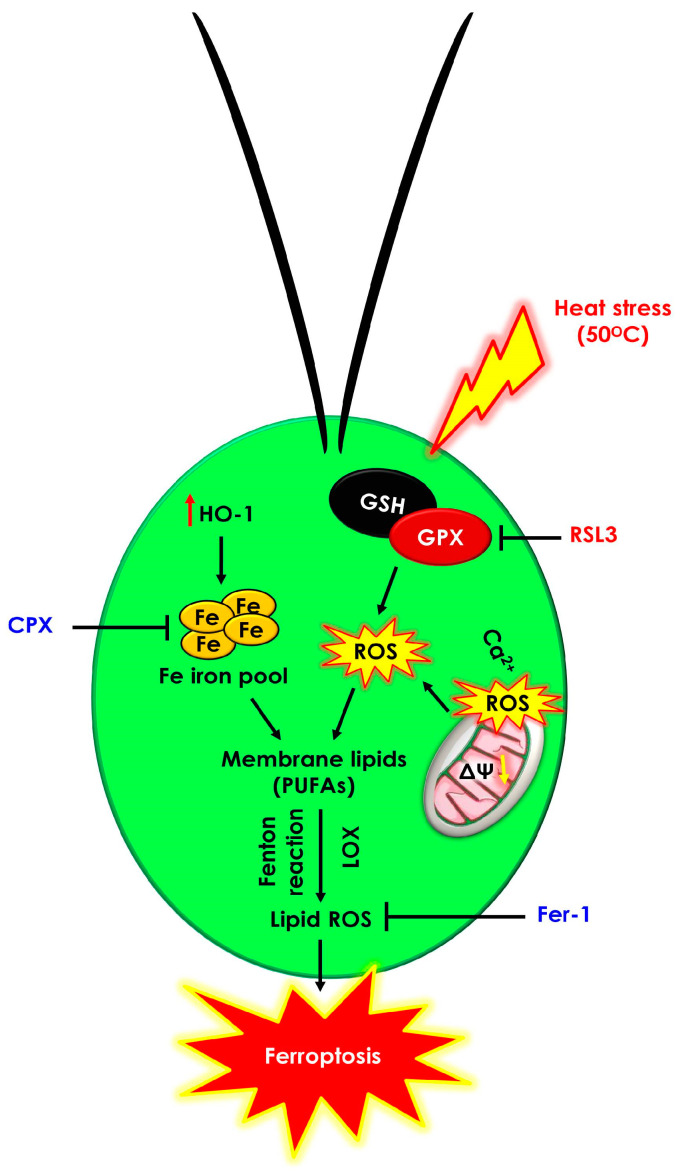
Proposed mechanism of ferroptotic cell death pathway induced by heat shock (50 °C) in the unicellular algae *C. reinhardtii*. Early depletion of glutathione and GPX5 inactivation lead to intracellular calcium, lipid ROS, and Fe^2+^ accumulation. Ferroptosis inhibitors (Fer-1 and CPX) significantly prevented the lipid-ROS- and Fe^2+^-dependent cell death induced by the RSL3 treatment and heat shock (50 °C).

## Data Availability

Not applicable.

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
