# Peer review of "Lipid ROS- and Iron-Dependent Ferroptotic Cell Death in Unicellular Algae Chlamydomonas reinhardtii"

_cells, 2023, doi:10.3390/cells12040553_

Round 1
Reviewer 1 Report
The manuscript by Srinivasan et al reports a series of physiological, biochemical and gene expression changes that take place in Chlamydomonas reinhardtii cells in response to a ferroptosis inducer as well as a 10-min heat stress at 50C in the presence of ferroptosis inhibitors. Based on the data presented, the authors conclude that heat stress induces a ferroptosis-like cell death in C. reinhardtii.
The research approach is based on similar studies in cancer cells as well as the plant Arabidopsis thaliana. However, this reviewer feels that some aspects need clarification (see below); also, the lack of other specific information/data (see below) makes the interpretation/conclusion – in their current form, less convincing.
Required clarification/data/information:
In addition to the ferroptosis hallmarks mentioned and investigated, ferroptosis is distinct from apoptosis-like cell death in terms of not being associated with mitochondrial cytochrome c release, caspase activation, and chromatin fragmentation. These (or some) features have not been observed in RLS-treated cells in plant and animal cells. Also, in A. thaliana, although a 55°C heat stress–induced death requires caspase-like activity, it does not result in DNA fragmentation. Why were these changes not investigated in C. reinhardtii? Especially as, in fact, the same heat-stress conditions as used in this study have been reported to actually induce DNA fragmentation/laddering (an important hallmark of apoptosis) in C. reinhardtii (doi:10.1016/j.febslet.2006.04.044 ). How does this latter aspect of heat-stress induced death in C. reinhardtii fit with the conclusion that heat-stress induced death in C. reinhardtii is in fact ferroptosis-like since chromatin fragmentation is not supposed to take place during ferroptosis (but is a hallmark of apoptosis)?
Figure 1B: it is not clear how the distinction between live and dead cells was made. Sytox 9 stains both live and dead cells; and PI stains only dead cells. Thus, dead cells will stain with both dyes; while live cells with only Sytox 9. So, green cells will fluoresce in green, and dead cells will fluoresce in BOTH green and red. If the pictures shown in Figure 1B are overlaps of both captured images, dead cells should appear orange (ie, both green and red). The legend states that live cells are in green, and dead cells are in red; how about the yellow/orange cells – how were they counted? In the cells treated with ferroptosis inhibitors there are very few (if any) green (live) cells (see the real green fluorescence indicative of live cells in the Control); were the yellow cells counted as live? Showing the images captured with the individual filters would be more informative; and counting should be done with those filters (live cells would represent the number of green cells minus the number of red cells). Alternatively, yellow/orange cells should be counted as dead. How long after the heat stress was the viability assessed?
Why 50C for 10 min? The C. reinhardtii studies mentioned as providing previous evidence for ferroptosis in terms of heat stress-induced ROS and lipid peroxidation used 40C or 42C for 60 min (and in Figure 6, the heat-stress indicated is 42C not 50C; although the legend states 50C).
Describe/mention the mechanism of action (when known) for the ferroptosis inducers and inhibitors (since they have been developed for animal cells)
Abstract: “indicating that eukaryotic cells share defined features of cell death pathways via convergent evolution.”….???? Why convergent evolution? If anything, the shared features indicate old common ancestry (also see the suggestion of in https://doi.org/10.1083/jcb.201605110: “The similarity of ferroptosis in animal cells and ferroptosis-like death in plants suggests that oxidative, iron-dependent cell death programs may be evolutionarily ancient)
Introduction:
Line 47: “Cell death is a vital process”; this might be ok for multicellular organisms, but for single cell organisms cell death can’t be “vital”… it is lethal…
Line 54: “Among these cell deaths, ferroptosis, a new form cell death was observed in prokaryotes and eukaryotes “ – which eukaryotes? Include references.
Editorial recommendations
- The text could benefit from a thorough grammar and spell check.
Author Response
Response to Reviewer 1 Comments
Dear Editor,
We would like to thank you and the reviewers for their interest in our manuscript and appreciate the effort and time they took in reading and commenting on it. We have revised our manuscript based on the reviewer’s comments to improve its present quality. Changes/justifications to their comments and suggestions are given below. The minor and major revisions suggested by reviewers were included in the revised manuscript with track changes.
Reviewer comments
The manuscript by Srinivasan et al reports a series of physiological, biochemical and gene expression changes that take place in Chlamydomonas reinhardtii cells in response to a ferroptosis inducer as well as a 10-min heat stress at 50C in the presence of ferroptosis inhibitors. Based on the data presented, the authors conclude that heat stress induces a ferroptosis-like cell death in C. reinhardtii.
The research approach is based on similar studies in cancer cells as well as the plant Arabidopsis thaliana. However, this reviewer feels that some aspects need clarification (see below); also, the lack of other specific information/data (see below) makes the interpretation/conclusion – in their current form, less convincing.
Required clarification/data/information:
Comment 1: In addition to the ferroptosis hallmarks mentioned and investigated, ferroptosis is distinct from apoptosis-like cell death in terms of not being associated with mitochondrial cytochrome c release, caspase activation, and chromatin fragmentation. These (or some) features have not been observed in RLS-treated cells in plant and animal cells. Also, in A. thaliana, although a 55°C heat stress–induced death requires caspase-like activity, it does not result in DNA fragmentation. Why were these changes not investigated in C. reinhardtii? Especially as, in fact, the same heat-stress conditions as used in this study have been reported to actually induce DNA fragmentation/laddering (an important hallmark of apoptosis) in C. reinhardtii (doi:10.1016/j.febslet.2006.04.044 ). How does this latter aspect of heat-stress induced death in C. reinhardtii fit with the conclusion that heat-stress induced death in C. reinhardtii is in fact ferroptosis-like since chromatin fragmentation is not supposed to take place during ferroptosis (but is a hallmark of apoptosis)?
Response 1: Thank you for your insights. As preliminary work, we also performed the effect of caspase inhibitor and DNA fragmentation on Chlamydomonas reinhardtii during heat shock and RSL3 treatment. To determine caspase-like activity in Chlamydomonas during RSL3 and heat shock treatment, we investigated the effect of pre-incubation with the reversible caspase-3 inhibitor (CHO). We observed that pre-incubation with CHO significantly reduced the percentage of cell death induced by RSL3 and heat shock. This suggested that a caspase-like activity was involved in the pathway triggered by RSL3 and heat shock. To assess the apoptosis-like cell death, we investigated DNA fragmentation assay by agarose gel electrophoresis. No DNA fragmentation was observed in the Chlamydomonas cells for 12 h after heat shock and RSL3 treatment (provided as Figure S2. in supplementary data). However, Nedelcu et al. 2006, observed that DNA fragmentation in Chlamydomonas during heat treatment at 50ᴼC for 10 min. Unfortunately, they failed to show the DNA fragmentation for different time points after heat treatment. This might be due to the majority of cells had lost their viability at the last stage of cell death. Our findings suggested that DNA fragmentation might not occur until for 12 h after heat shock, when cell death has already been detected in cells. These results are consistent with previous findings of Distefano et al. 2017; Hajdinak et al. 2019 in Arabidopsis thaliana cells.
References
- Distéfano, A. M., Martin, M. V., Córdoba, J. P., Bellido, A. M., D’Ippólito, S., Colman, S. L., & Pagnussat, G. C. (2017). Heat stress induces ferroptosis-like cell death in plants. Journal of Cell Biology, 216(2), 463-476.
- Hajdinák, P., Czobor, Á., & Szarka, A. (2019). The potential role of acrolein in plant ferroptosis-like cell death. PLoS One, 14(12), e0227278.
Comment 2: Figure 1B: it is not clear how the distinction between live and dead cells was made. Sytox 9 stains both live and dead cells; and PI stains only dead cells. Thus, dead cells will stain with both dyes; while live cells with only Sytox 9. So, green cells will fluoresce in green, and dead cells will fluoresce in BOTH green and red. If the pictures shown in Figure 1B are overlaps of both captured images, dead cells should appear orange (ie, both green and red). The legend states that live cells are in green, and dead cells are in red; how about the yellow/orange cells – how were they counted? In the cells treated with ferroptosis inhibitors there are very few (if any) green (live) cells (see the real green fluorescence indicative of live cells in the Control); were the yellow cells counted as live? Showing the images captured with the individual filters would be more informative; and counting should be done with those filters (live cells would represent the number of green cells minus the number of red cells). Alternatively, yellow/orange cells should be counted as dead. How long after the heat stress was the viability assessed?
Response 2: Thank you for your suggestion. In our study, we also followed the rule of considering yellow/orange stained cells as dead cells. After heat shock at 50ᴼC, we performed the cell death kinetics of Chlamydomonas cells for 24 h.
Comment 3: Why 50C for 10 min? The C. reinhardtii studies mentioned as providing previous evidence for ferroptosis in terms of heat stress-induced ROS and lipid peroxidation used 40C or 42C for 60 min (and in Figure 6, the heat-stress indicated is 42C not 50C; although the legend states 50C).
Response 3: In an earlier research, either 42ᴼC for 60 min or a more severe heat stimulus 50ᴼC for 10 min, has been reported to produce similar results in Chlamydomonas cells (Druand et al. 2014). Similarly, other studies have also proved the acute shock 50ᴼC for 10 min is sufficient to induced programmed cell death in Chlamydomonas reinhardtii (Nedelcu et al. 2006; Druand et al. 2011). For the figure 6, our sincere apologies for mistake, we have changed the typo error 42ᴼC to 50ᴼC.
References
- Durand, P. M., Choudhury, R., Rashidi, A., & Michod, R. E. (2014). Programmed death in a unicellular organism has species-specific fitness effects. Biology Letters, 10(2), 20131088.
- Durand, P. M., Rashidi, A., & Michod, R. E. (2011). How an organism dies affects the fitness of its neighbors. The American Naturalist, 177(2), 224-232.
- Nedelcu, A. M. (2006). Evidence for p53-like-mediated stress responses in green algae. FEBS letters, 580(13), 3013-3017.
Comment 4: Describe/mention the mechanism of action (when known) for the ferroptosis inducers and inhibitors (since they have been developed for animal cells)
Response 4: Thank you for your suggestion, we have mentioned the property of inhibitors (lipophilic antioxidant – Ferrostatin-1 and Iron chelator – CPX) and mechanism of ferrotopsis inducer (RSL3) details in the Introduction section. Moreover, we have also provided the currently accepted mechanism of ferroptosis inducer and inhibitors role in Chlamydomonas cells (see in Figure. 6)
Comment 5: Abstract: “indicating that eukaryotic cells share defined features of cell death pathways via convergent evolution.”….???? Why convergent evolution? If anything, the shared features indicate old common ancestry (also see the suggestion of in https://doi.org/10.1083/jcb.201605110: “The similarity of ferroptosis in animal cells and ferroptosis-like death in plants suggests that oxidative, iron-dependent cell death programs may be evolutionarily ancient)
Response 5: Thank you for your suggestion, we have edited the sentence as recommended by reviewer
Comment 6: Line 47: “Cell death is a vital process”; this might be ok for multicellular organisms, but for single cell organisms cell death can’t be “vital”… it is lethal…
Response 6: Thank you. We have changed the sentence as “Cell death is a fundamental biological process…”
Comment 7: Line 54: “Among these cell deaths, ferroptosis, a new form cell death was observed in prokaryotes and eukaryotes “ – which eukaryotes? Include references.
Response 7: Thank you for your recommendation. We have edited the sentence and added reference.
Comment 8: Editorial recommendations
- The text could benefit from a thorough grammar and spell check.
Response 8: We have checked the grammar and spell check throughout the manuscript using native English speaker according to editor recommendations

Reviewer 2 Report
See the attached file.

Author Response
Response to Reviewer 2 Comments
Dear Editor,
We would like to thank you and the reviewer for their interest in our manuscript and appreciate the effort and time they took in reading and commenting on it. We have revised our manuscript based on the reviewer’s comments to improve its present quality. Changes/justifications to their comments and suggestions are given below. The minor and major revisions suggested by reviewers were included in the revised manuscript with track changes.
Reviewer comments
General:
Comment 1: ▪ Please check for grammar and any typos once again. Proof-read it multiple times by different people.
Response 1: Thank you. We have checked grammar and typo errors throughout the manuscript.
Comment 2: ▪ Some figures are misaligned. Especially Fig2: you can just arrange the parts vertically that way you can make the graphs bigger than original.
Response 2: Thank you for suggestion, we have rearranged the figure.2 in vertical position as recommended by reviewer.
Introduction
Comment 3: ▪ Line 60: need to clarify what RSL3 and HSF1-HSPB1 stands for. Just a general note please before using any abbreviation write what it stands for in full or just mention one line about it. Just stating the abbreviations doesn’t help the reader.
Response 3: Apologies for our mistake, we have provided the abbreviations for RSL3 and HSF1-HSPB1 in the Introduction section
Comment 4: ▪ Line 60-61: all the molecules mentioned in line 60 together or co-operatively induce all the pathways stated in line 61 or they specifically trigger some or each of them. Please clarify accordingly.
Response 4: Sorry for confusion, we have rectified and rephrased the sentence completely.
Comment 5: ▪ Line 67: the word “enhanced” might be inappropriate here. I am guessing you mean to say presence of iron intensify or escalate the ferroptosis cell death. If yes, please I suggest using a different word instead of “enhance”. Enhance seems to be vague in this respect.
Response 5: Thank you for your suggestion, we have changed the word to “triggered” in the manuscript.
Comment 6: ▪ Line 72-73: only genes can show differential expression and not proteins. Please revise accordingly.
Response 6: Thank you. We have changed the word proteins into genes.
Comment 7: ▪ Line 81-83: is it possible that the defined characteristics described here can be attributed to any other factors or processes and not just ferroptosis cell death? If yes, then those need to be clarified and the scientific premise needs to be established more precisely.
Response 7: In microalgae, profound studies have been reported that GSH/GPX antioxidant system acts as first line defence mechanism during oxidative stress induced by environmental stimuli (Pikula et al. 2019). Surprisingly, during heat stress condition, GSH and GPX level was significantly reduced and results in ROS accumulation and lipid peroxidation in Chlamydomonas cells as described in previous reports. However, exact mechanism for depletion of GSH/GPX system and lipid-ROS accumulation during heat stress in microalgae remains unknown. This is first report, we observed that heat stress causes early depletion of glutathione and GPX5 inactivation leads to intracellular calcium, lipid ROS and Fe2+ iron accumulation by regulation of heme oxygenase-1 in microalgae. These results were consistent with previous findings of ferroptosis-like cell death in mammalian cells, plants and bacteria (Dixon et al. 2012; Distefano et al. 2017; Hajdinák et al. 2019; Aguilera et al. 2021)
References
- Pikula, K. S., Zakharenko, A. M., Aruoja, V., Golokhvast, K. S., & Tsatsakis, A. M. (2019). Oxidative stress and its biomarkers in microalgal ecotoxicology. Current Opinion in Toxicology, 13, 8-15.
- Dixon, S. J., Lemberg, K. M., Lamprecht, M. R., Skouta, R., Zaitsev, E. M., Gleason, C. E., & Stockwell, B. R. (2012). Ferroptosis: an iron-dependent form of nonapoptotic cell death. Cell, 149(5), 1060-1072.
- Distéfano, A. M., Martin, M. V., Córdoba, J. P., Bellido, A. M., D’Ippólito, S., Colman, S. L., & Pagnussat, G. C. (2017). Heat stress induces ferroptosis-like cell death in plants. Journal of Cell Biology, 216(2), 463-476.
- Hajdinák, P., Czobor, Á., & Szarka, A. (2019). The potential role of acrolein in plant ferroptosis-like cell death. PLoS One, 14(12), e0227278.
- Aguilera, A., Berdun, F., Bartoli, C., Steelheart, C., Alegre, M., Bayir, H., & Martin, M. V. (2021). C-ferroptosis is an iron-dependent form of regulated cell death in cyanobacteria. Journal of Cell Biology, 221(2), e201911005.
▪ Some outstanding questions I have:
Comment 8: o How prevalent and relevant is the heat stress under dark conditions for these algae strain in natural environment?
Response 8: We would like to thank the author for his valuable insight. We will definitely consider this in our ongoing and future studies.
Comment 9: o Does this cell death happen in natural environment? If yes, then how frequent is it to study the mechanisms?
Response 9: Thank you for your comment. Although the chances of such phenomenon in nature are not wide, such studies can help to understand the survivability or adaptations of organisms in ecological niches such as hot springs. Also on a broader view studying processes like ferroptosis would help to understand changes occurring in cells due to climate change.
Comment 10: o And lastly, why do we care? (See the next point for more clarification)
▪ Major comment for Introduction:
o Additionally, it’s not very exciting just to figure out the mechanism of cell death although I agree it’s important but in my view end of an introduction should answer or give perspective into why knowing this knowledge is important in future. Also, why is it important to know for Chlamydomonas. I would suggest revising.
Response 10: We would like to thank the reviewer for his insight. We have modified our introduction to address this point.
Material and Methods
Comment 11: ▪ For cell death assay: how many cells were counted in total for one measurement? How is Cell death % calculated. This should be added in the Mat & Mtd section.
Response 11: we have included the method followed to calculate the percentage of cell death in the material and methods section.
Comment 12: ▪ Line 153: Chlamydomonas needs to be italicized
Response 12: Thank you, we have changed Chlamydomonas word italicized
Comment 13: ▪ Most assays are written in very specific detail which is great! Keep the good work!
Response 13: Thank you for you appreciation
Results and Discussion
Comment 14: ▪ Figure S1: Is there an explanation for the rapid spike in cell death on heat shock between 3-6 hrs?
Response 14: ROS accumulation (1h and 3 h) and lipid peroxidation was found to be higher during early stage of cell death. Moreover, antioxidants (GSH and ascorbate) and Fe2+ accumulation also significantly increased in Chlamydomonas after 4h heat shock treatment. This might attribute the rapid spike in cell death during period of 3-6 h. We also observed that ROS kinetics was significantly reduced after 6h in Chlamydomonas cells during preliminary study.
Comment 15: ▪ Are there any known factors that can cause other form of cell death? I think it will be worth to show that iron-specific cell death inhibitors do not work when a different stressor is used. I think it’s an important control to show specificity of these stressors.
Response 15: In our preliminary work, we observed that both the ferroptosis inhibitors CPX and Fer-1 failed to prevent the cell death in response to osmotic (KCl) and oxidative stress (H2O2). Moreover, previous studies have reported that apoptosis-like cell death (i.e., DNA fragmentation) in Chlamydomonas reinhardtii under osmotic and oxidative stress (Vavilala et al. 2015; Vavilala et al. 2016). In our study, heat shock 50ᴼC significantly induced the caspase-like activity but does not result in DNA fragmentation in Chlamydomonas (See Fig. S2 in supplementary data provided as recommended by Reviewer 1)
References
- Vavilala, S. L., Gawde, K. K., Sinha, M., & D’Souza, J. S. (2015). Programmed cell death is induced by hydrogen peroxide but not by excessive ionic stress of sodium chloride in the unicellular green alga Chlamydomonas reinhardtii. European Journal of Phycology, 50(4), 422-438.
- Vavilala, S. L., Sinha, M., Gawde, K. K., Shirolikar, S. M., & D'Souza, J. S. (2016). KCl induces a caspase-independent programmed cell death in the unicellular green chlorophyte Chlamydomonas reinhardtii (Chlorophyceae). Phycologia, 55(4), 378-392.
Comment 16: ▪ The statistical significance represented here is a bit confusing. I honestly feel that comparison between control and for example RSL3 is okay but RSL3 + inhibitors need to be compared to the RSL3 treatment. It seems like the measurements are compared to control for their significance which in my view is not a fair comparison.
Response 16: Thank you for suggestion; we have included the recommended changes in the statistical analysis.
Comment 17: ▪ Is there any other way these ferroptotic markers can be induced? Pre-treatment with inhibitors fail short to abolish the cell death to the level of control. What could be the causes for this observation? then it seems iron-dependent cell death is probably not the only cause for the phenotype seen. Is there a possibility that two different mechanisms are working additively or independently?
Response 17: The nature of inhibitors varies (competitive, non-competitive or uncompetitive) and complete termination of any cellular process at a given point of time is practically not possible as there are dynamic reactions happening inside the cell. Also, circumstances such as environmental stress or targeted protein structure can also influence the inhibitors but the effect is often negligible (similar results observed in plants and cyanobacteria - Hajdinak et al. 2019; Li et al. 2022; Aguilera et al. 2021).
References
- Hajdinák, P., Czobor, Á., & Szarka, A. (2019). The potential role of acrolein in plant ferroptosis-like cell death. PLoS One, 14(12), e0227278.
- Li, J., Chen, S., Huang, J., Chen, H., Chen, Z., & Wen, Y. (2021). New Target in an Old Enemy: Herbicide (R)-Dichlorprop Induces Ferroptosis-like Death in Plants. Journal of Agricultural and Food Chemistry, 69(27), 7554-7564.
- Aguilera, A., Berdun, F., Bartoli, C., Steelheart, C., Alegre, M., Bayir, H., & Martin, M. V. (2021). C-ferroptosis is an iron-dependent form of regulated cell death in cyanobacteria. Journal of Cell Biology, 221(2), e201911005.
Comment 18: ▪ Line 262-263: the sentence doesn’t sound correct. Please revise the wording.
Response 18: Thank you. We have rephrased the sentence as recommended by reviewer.
Comment 19: ▪ Line 264: typo in word shock
Response 19: Thank you. We have changed the typo error.
Comment 20: ▪ Figure 6: is there any reason the heat stress here is 42C than 50C? or is it a typo? I love the model figure, just need to increase font, use little less bright or neon colors and make it more professional looking. I suggest using biorender.
Response 20: Apologies for our mistake. It is typo error; we have changed the heat stress to 50ᴼC. We have improved the figure color and increased font size.
▪ Main comment for results:
Comment 21: o How can you say that heat stress causes only ferroptotic cell death? The results clearly show some other process is also happening which can also attribute to all the markers or assays depicted here. The specificity that its only iron-dependent cell death is very indistinct.
Response 21: Ferroptosis is a non-apoptotic and oxidative damage-related RCD (Dixon et al. 2012), mainly driven by iron accumulation and lipid peroxidation. In case of other types of cell death, DNA fragmentation and caspase activity are observed as hallmarks. We only observed caspase-like activity but no DNA fragmentation in the present study (Figure S2, see supplementary data). These results are consistent with previous findings of Distefano et al. 2017; Hajdinak et al. 2019 studying ferroptosis in Arabidopsis thaliana cells.
References
- Dixon, S. J., Lemberg, K. M., Lamprecht, M. R., Skouta, R., Zaitsev, E. M., Gleason, C. E., & Stockwell, B. R. (2012). Ferroptosis: an iron-dependent form of nonapoptotic cell death. Cell, 149(5), 1060-1072.
- Distéfano, A. M., Martin, M. V., Córdoba, J. P., Bellido, A. M., D’Ippólito, S., Colman, S. L., & Pagnussat, G. C. (2017). Heat stress induces ferroptosis-like cell death in plants. Journal of Cell Biology, 216(2), 463-476.
- Hajdinák, P., Czobor, Á., & Szarka, A. (2019). The potential role of acrolein in plant ferroptosis-like cell death. PLoS One, 14(12), e0227278.
Comment 22: o Does heat stress (50C) specifically inhibits GSH or GPX that none of the other stressors can which eventually lead to iron-dependent cell death? If not, then the main or committed step is not very distinct here. If its upregulation of HO1 coding gene, then it needs to be the highlight of the paper.
Response 22: In ferroptotic pathway, GSH/GPX system is an initial regulatory step (Dixon et al. 2021) and followed by ROS and lipid peroxidation in cells (Dixon et al. 2021). Similarly, Distefano et al. and Aguilera et al. 2021 were observed that depletion of GSH/GPX system leads to ROS and lipid peroxidation in plant and bacteria during heat stress. Moreover, exogenous addition of herbicides also causes similar effects in the model plant, Arabidopis thaliana (Hajdinak et al. 2019; Li et al. 2021). Similarly, our findings also corroborates with these previous reports. Second, HO-1 (also heat shock protein), is a stress-inducible enzyme that plays important roles in iron homeostasis which causes detrimental effects via ferroptosis induction during heat stress (Liu et al. 2022; Yang and Stockwell, 2016).
References
- Distéfano, A. M., Martin, M. V., Córdoba, J. P., Bellido, A. M., D’Ippólito, S., Colman, S. L., & Pagnussat, G. C. (2017). Heat stress induces ferroptosis-like cell death in plants. Journal of Cell Biology, 216(2), 463-476.
- Aguilera, A., Berdun, F., Bartoli, C., Steelheart, C., Alegre, M., Bayir, H., & Martin, M. V. (2021). C-ferroptosis is an iron-dependent form of regulated cell death in cyanobacteria. Journal of Cell Biology, 221(2), e201911005.
- Dixon, S. J., Lemberg, K. M., Lamprecht, M. R., Skouta, R., Zaitsev, E. M., Gleason, C. E., & Stockwell, B. R. (2012). Ferroptosis: an iron-dependent form of nonapoptotic cell death. Cell, 149(5), 1060-1072.
- Hajdinák, P., Czobor, Á., & Szarka, A. (2019). The potential role of acrolein in plant ferroptosis-like cell death. PLoS One, 14(12), e0227278.
- Li, J., Chen, S., Huang, J., Chen, H., Chen, Z., & Wen, Y. (2021). New Target in an Old Enemy: Herbicide (R)-Dichlorprop Induces Ferroptosis-like Death in Plants. Journal of Agricultural and Food Chemistry, 69(27), 7554-7564.
- Liu, Y., Zhou, L., Xu, Y., Li, K., Zhao, Y., Qiao, H., & Zhao, J. (2022). Heat Shock Proteins and Ferroptosis. Frontiers in Cell and Developmental Biology, 771.
- Yang, W. S., & Stockwell, B. R. (2016). Ferroptosis: death by lipid peroxidation. Trends in cell biology, 26(3), 165-176.
Comment 23: o Additionally, as I had stated earlier, there seems like some processes occurring that prevents full abolishment of the various markers tested here. This needs to be addressed either by digging in literature or experimentally.
Response 23: As explained in an earlier comment this phenomenon occurs due to changes in the environment or in the cellular processes and is often negligible.
Comment 24: Conclusion: The conclusion is short and concise but again the scope is missing. What can a researcher do with this knowledge? How can it be applied? It needs to justify the importance of research.
Response 24: Thank you for your comment. We have modified the conclusion part according to the advice of the reviewer.

Round 2
Reviewer 1 Report
Additional suggested revisions:
- Since distinct Methods as Results & Discussion sections on caspase activity and DNA fragmentation are now included in the manuscript, the data should also be included in the main text; as it is now, the Results section on these 2 assays has no data shown…
- Also, the section on DNA fragmentation should include a discussion of previous studies (Durand et al 2014; Nedelcu 2006) that show the induction of DNA laddering; for instance, Durand et al used both 42C for 1 h and 50C for 10 min and showed the DNA 14–16 h post-stress. As it is now, the lack of DNA fragmentation is presented as consistent with findings in Arabidopsis, but there is no mention (and discussion) of studies showing DNA fragmentation in Chlamy, and why no fragmentation was observed in the current study (including the fact that very small amounts of DNA were loaded on the gel in the current study, such that the bands corresponding to the DNA fragments of smaller size are harder to detect; see, for instance, corresponding figures in Durand et al 2014 and Nedelcu 2006)
- Generally, the reader would benefit from a short discussion of evidence of apoptosis-like cell death in Chlamydomonas to assess the relevance of ferroptosis in the context of cell death in Chlamydomonas (there is a large body of research and literature on cell death in Chlamydomonas). As it is now, the reader is left with the impression that ferroptosis is the only mechanism of regulated cell death in Chlamydomonas
- Define “system Xc”
- There are still grammar issues, especially in the newly added sections; eg:
o Among these cell deaths, ferroptosis, a new form cell death was observed in organisms which differs physiologically, biochemically and genetically from apoptosis, necrosis and autophagic cell death
o Studying the cellular processes such as ferroptosis can help us understand earlier any unsolved phenomenon observed in algae, provide insight into its biotechnological implications
o percentage of dead cells (PI positive cells) to the total number of cells
o cells were collected by centrifuge at
o the cells was washed
Author Response
Response to Reviewer 1 comments
Dear Editor,
We appreciate you and the reviewers for your precious time in reviewing our paper and providing valuable comments. It was your valuable and insightful comments that led to possible improvements in the current version. The authors have carefully considered the comments and tried our best to address every one of them. The minor revisions suggested by reviewers were included in the revised manuscript with track changes.
Reviewer comments
Comment 1: Since distinct Methods as Results & Discussion sections on caspase activity and DNA fragmentation are now included in the manuscript, the data should also be included in the main text; as it is now, the Results section on these 2 assays has no data shown…
Response 1: Thank you very much for the insight. Since ferroptosis is a non-apoptotic cell death, caspase activity and DNA fragmentation assay was preliminarily performed only to confirm that heat shock treatment (50 ᴼC) induces caspase activity and not DNA fragmentation in Chlamydomonas cells. Therefore, we have included these data in the supplementary file as they are not one of our key results (as Figure S2).
Comment 2: Also, the section on DNA fragmentation should include a discussion of previous studies (Durand et al 2014; Nedelcu 2006) that show the induction of DNA laddering; for instance, Durand et al used both 42C for 1 h and 50C for 10 min and showed the DNA 14–16 h post-stress. As it is now, the lack of DNA fragmentation is presented as consistent with findings in Arabidopsis, but there is no mention (and discussion) of studies showing DNA fragmentation in Chlamy, and why no fragmentation was observed in the current study (including the fact that very small amounts of DNA were loaded on the gel in the current study, such that the bands corresponding to the DNA fragments of smaller size are harder to detect; see, for instance, corresponding figures in Durand et al 2014 and Nedelcu 2006).
Response 2: Thank for your suggestion, we have discussed the results with previous studies and revised accordingly. In case of faint/low intensity bands we observed similar results (in terms of band intensity) even when slightly higher concentration of DNA used which may be due to technical problems with the gel documentation instrument/processing software. Since the results were evident and confirmed we continued with our experiments.
Comment 3: Generally, the reader would benefit from a short discussion of evidence of apoptosis-like cell death in Chlamydomonas to assess the relevance of ferroptosis in the context of cell death in Chlamydomonas (there is a large body of research and literature on cell death in Chlamydomonas). As it is now, the reader is left with the impression that ferroptosis is the only mechanism of regulated cell death in Chlamydomonas.
Response 3: Thank you for suggestion, we have revised accordingly
Comment 4: Define “system Xc”
Response 4: we have included the definition for system xc-
There are still grammar issues, especially in the newly added sections; eg:
Comment 5: Among these cell deaths, ferroptosis, a new form cell death was observed in organisms which differs physiologically, biochemically and genetically from apoptosis, necrosis and autophagic cell death.
Response 5: Revised accordingly
Comment 6: Studying the cellular processes such as ferroptosis can help us understand earlier any unsolved phenomenon observed in algae, provide insight into its biotechnological implications
Response 6: Revised accordingly
Comment 7: percentage of dead cells (PI positive cells) to the total number of cells
Response 7: Revised accordingly
Comment 8: cells were collected by centrifuge at
Response 8: Revised accordingly
Comment 9: the cells was washed
Response 9: Revised accordingly

Reviewer 2 Report
NA
Author Response
Response to Reviewer 2 comments
Dear Editor,
We appreciate you and the reviewers for your precious time in reviewing our paper and providing valuable comments. It was your valuable and insightful comments that led to possible improvements in the current version. We have revised the fine/minor errors throughout the manuscript.
Comments: NA
Thanking you,
Yours Sincerely
Dr. Mohandass Ramya
